# Linear Weight Interpolation Leads to Transient Performance Gains

**Gaurav Iyer**                                        *gaurav.iyer@mila.quebec*
*McGill University*
*Mila - Quebec AI Institute*

**Gintare Karolina Dziugaite**                         *gkdz@google.com*
*Google DeepMind*
*McGill University*
*Mila - Quebec AI Institute*

**David Rolnick**                                       *drolnick@mila.quebec*
*McGill University*
*Mila - Quebec AI Institute*

**Reviewed on OpenReview:** *https://openreview.net/forum?id=XGAdBXlFcj*

## Abstract

We train copies of a neural network on different sets of SGD noise and find that linearly interpolating their weights can, remarkably, produce networks that perform significantly better than the original networks. However, such interpolated networks consistently end up in unfavorable regions of the optimization landscape: with further training, their performance fails to improve or degrades, effectively undoing the performance gained from the interpolation. We identify two quantities that impact an interpolated network's performance and relate our observations to linear mode connectivity. Finally, we investigate this phenomenon from the lens of example importance and find that performance improves and degrades almost exclusively on the harder subsets of the training data, while performance is stable on the easier subsets. Our work represents a step towards a better understanding of neural network loss landscapes and weight interpolation in deep learning.

## 1 Introduction

Linear interpolations of neural network weights are of considerable interest in modern deep learning, for both theoretical and practical purposes (Singh & Jaggi, 2020; Li et al., 2023; Neyshabur et al., 2020). They have aided our understanding of training dynamics and loss landscapes (Frankle et al., 2020; Sharma et al., 2024; Paul et al., 2022a), and can improve performance at convergence (Izmailov et al., 2018; Matena & Raffel, 2022; Wortsman et al., 2022).

Much of the prior work on this topic has focused on linear interpolations of network weights during late-stage training or at convergence (Wortsman et al., 2022), or interpolating SGD iterates of a network along a single training trajectory (Zhang et al., 2019). However, our understanding of the properties of such networks and their optimization remains limited in more general settings. The observations and investigations presented here attempt to address this gap and improve our understanding of how linear interpolations evolve in the context of SGD noise throughout the training process.

We train copies of networks on different sets of SGD noise (i.e data order and augmentation) on standard vision tasks. More specifically, we consider a network initialization that is trained for a small number of iterations $k$, after which its copies A and B are trained on different sets of SGD noise for $s$ epochs.

With this context, our main findings and contributions are summarized as follows:

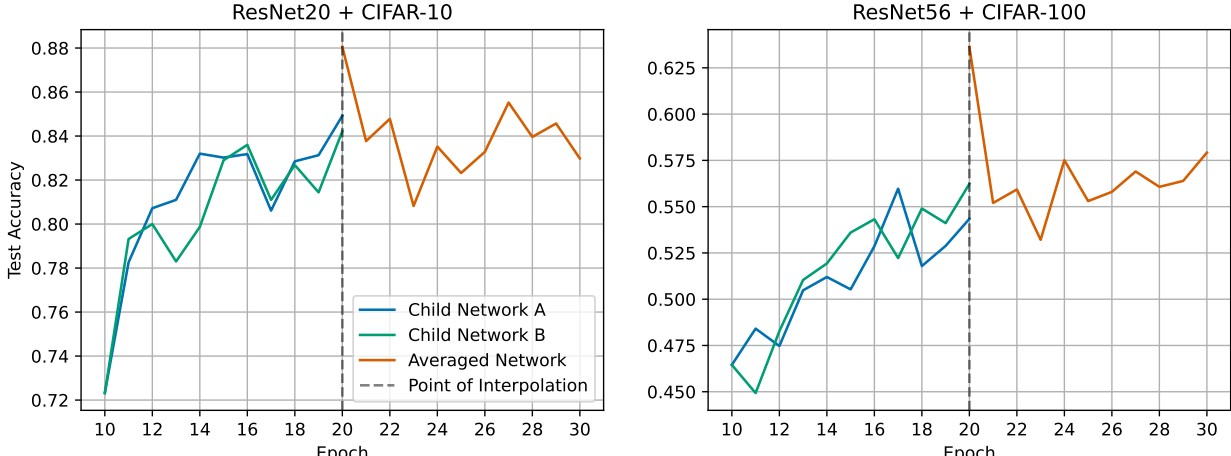

Figure 1: We train two copies of a network on different sets of SGD noise, then average their weights and continue training on the resulting network. We find that test accuracy shoots up upon interpolation, but then precipitously drops and improvement stalls. The network initialization was trained for $k = 10$ epochs before being cloned into "child networks" A and B. A and B were then trained on different SGD trajectories for $s = 10$ epochs, before being averaged and trained further for 10 epochs.

- We show that the performance of the network derived by interpolating A and B (referred to as the **_interpolant_** henceforth) varies significantly throughout the optimization process. Specifically, given $k$, the interpolant can display better accuracy than A or B on the training and test set, if the value of $s$ is small enough. Later in training (i.e. as $s$ becomes larger), interpolant performance becomes significantly worse.

- We attempt to train interpolants derived early in training and find that the improvement in performance is seemingly transient – while they display better immediate performance, they either fail to improve or degrade over the next few epochs of training. Empirically, we find that standard learning rate schedule adjustments effectively provide no speed-up in optimization. We hypothesize that first-order optimization methods may be unsuitable for training interpolants and maintaining the accuracy boost.

  We also note that trained interpolated networks perform just as well as standard networks when trained to convergence – by interpolating early, one can obtain models that perform better only early in training.

- We analyze this phenomenon through the lens of example importance for good generalization, and find that "important" examples (those with higher EL2N scores as defined in Paul et al. (2021)) contribute significantly to the boost *and* subsequent drop in performance. Furthermore, there is nearly no drop in performance on the less important examples – there can be no increase in performance in this case since networks perfectly learn these examples early in training. We hypothesize that our observations occur due to different subpopulations of the data becoming stable to SGD noise at different points in training, as noted by Paul et al. (2021).

## 2 Related Work

Network weight interpolation has attracted significant interest in recent work. Izmailov et al. (2018) introduces Stochastic Weight Averaging, where averaging SGD iterates along the same trajectory lead to better generalization. In a similar vein, Zhang et al. (2019) introduces the Lookahead optimizer, which interpolates a set of "fast weights" to update the actual weights of the network, leading to better training stability

and generalization. Wortsman et al. (2022) show that networks fine-tuned with different hyperparameter configurations often lie in the same loss basin, and that averaging them can lead to better robustness and performance. Ilharco et al. (2023) introduce the notion of "task vectors", which capture task-specific directions in weight space. Furthermore, they can be arithmetically combined with fine-tuned network weights and with each other to improve and/or degrade performance on specific tasks. Several other works support these observations (Matena & Raffel, 2022; Li et al., 2022; McMahan et al., 2017). In contrast to these settings, our work focuses on interpolants obtained in the earlier stages of training.

Our work is largely based on the framework provided by Frankle et al. (2020), which introduced linear mode connectivity – networks trained from the same initialization on different sets of SGD noise (after some short period of shared training) converge to the same linearly connected minimum. We build upon their work by investigating interpolants earlier in training, instead of at convergence. Mirzadeh et al. (2021) observe that networks starting with the same initialization are connected by linear, low-loss paths in the context of continual and multitask learning. Zhou et al. (2023) show that this notion of linear mode connectivity extends to layer-wise feature maps as well. Permutation symmetries have been shown to align networks so that they become linearly mode connected (Entezari et al., 2022; Ainsworth et al., 2023; Sharma et al., 2024). Garipov et al. (2018) and Draxler et al. (2018) show that networks trained from different initializations can be connected by non-linear low-loss paths.

Paul et al. (2021) propose the GraNd and EL2N scores to identify important examples early in training across different network architectures and training configurations. We make use of the EL2N score to analyze observations made in this work and relate it to example importance. Toneva et al. (2018) shows that deep networks learn "easy" data earlier in training and rarely forget this subset. Furthermore, these examples do not contribute significantly to final generalization performance. Our experiments support these findings – we find that interpolated network performance is most variable on examples that are learned late in training. Baldock et al. (2021) introduces the notion of effective prediction depth to empirically measure example importance and correlate it to the accuracy and speed of learning a given example.

## 3 Interpolating Weights Early in Training

### 3.1 Linear Mode Connectivity and Instability Analysis

We say that two neural networks are linearly connected if all networks along the linear path connecting the two trained networks in weight space have loss no larger than the end points (Frankle et al., 2020). The authors empirically demonstrate this through instability analysis, where two copies of a network are trained on different samples of SGD noise to convergence. If the two converged networks are linearly connected, the original network is said to be stable to SGD noise.

In our work, we cannot use instability directly – as we will soon see, we often encounter interpolants that perform better than the networks they were derived from, and therefore taking a maximum over performance errors will not suit our purposes. Instead, we consider a similar metric but pay attention to the maximum difference (increase or decrease) from the average error of the two networks that we are interpolating between. Unlike (Frankle et al., 2020), we track the interpolant's performance throughout training rather than only at convergence.

### 3.2 Interpolation Performance Throughout Training

Understanding the effect of linear interpolation early in training helps improve our understanding of linear mode connectivity since it is well-known that the early, noisy stage of training can heavily impact training dynamics and performance at later stages (Goyal et al., 2018; Jastrzebski et al., 2020; Gilmer et al., 2022; Paul et al., 2022b; Fort et al., 2020). Our analysis can also help tease apart the properties and structure of neural network loss landscapes at different stages in the training process.

We start with a network that is randomly initialized or trained for $k$ steps. Then, two copies of this network, A and B, are trained on different sets of SGD noise for $s$ epochs. Experimental details for the training setup can be found in Appendix A. We will show the results of our experiment on CIFAR-10 (Krizhevsky, 2009)

with ResNet20 networks (He et al., 2016) here (specifically Fig. 2 in this context) – additional results on different datasets and network architectures can be found in Appendix E and onward.

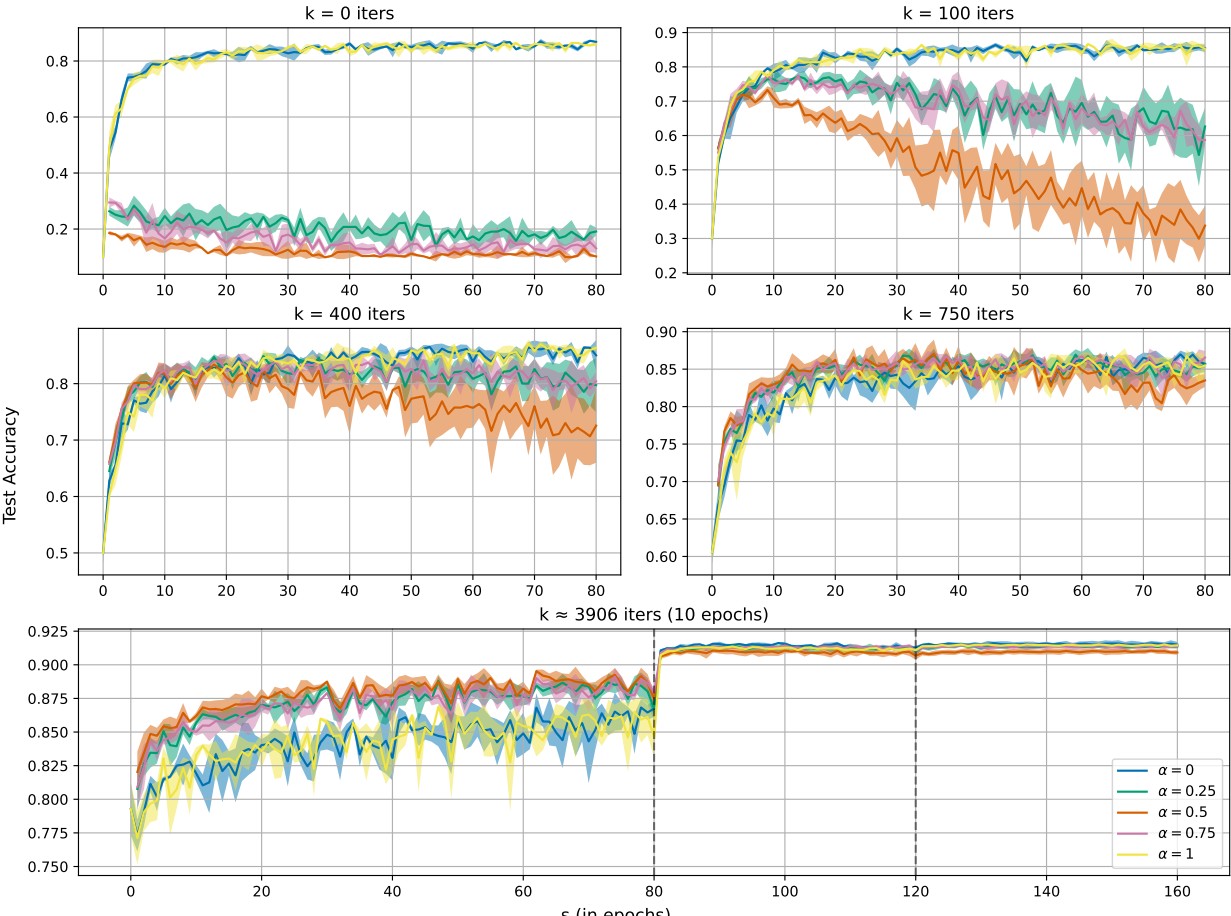

Figure 2: Progression of test performance of ResNet20 networks resulting from linear weight interpolation of child networks ($\alpha = 0$ and 1, where $\alpha$ is the interpolation weight) trained on different sets of SGD noise on CIFAR-10. The network initialization is trained for $k$ steps before being cloned into child networks, after which they are trained on different SGD trajectories for $s$ epochs. As $k$ increases, interpolants perform just as well or better than the child networks for an increasing number of epochs $s$ before their performance degrades. Eventually, interpolants perform well at all points in training until convergence, making the network "stable to SGD noise" à la Frankle et al. (2020). The vertical dotted black lines represent learning rate drops by a factor of 0.1, following the standard training scheme for this task. Note that the interpolants are not trained further after interpolation – their performance is simply recorded while child networks are trained on different sets of SGD noise. We repeat this experiment on 3 network initializations and provide error bars.

We observe that interpolant performance depends on both $k$ and $s$. When $k = 0$, i.e. copies of the network are made at initialization without any training, interpolants perform extremely poorly. However, when the value of $k$ is increased slightly, we make **two observations**:

(a) the interpolants slightly outperform networks A and B early in training, and

(b) the rate at which interpolant performance degrades becomes smaller.

Furthermore, as $k$ scales, both these observations become more dominant – interpolants outperform the original networks for a longer portion of the training period, and their performance degrades more slowly. Note that the interpolants are not trained further – their performance is simply noted while networks A and B continue training on separate SGD trajectories.

Once $k$ is sufficiently large, interpolant performance improves by more or less the same amount regardless of the exact value of $s$. This is important from a practical perspective – we can potentially improve the performance of a network by simply training two copies on different sets of SGD noise for a small number of steps and interpolating their weights. Based on the scale and nature of the task, this may be preferable to training a single network for multiple epochs. It can be reasoned that the performance of an interpolated network drops as $s$ increases since networks A and B grow further apart in L2 distance in weight space when trained through different SGD trajectories. The same observation applies to other notions of distance as well (see Appendix G of Frankle et al. (2020)).

Observation (b) is partly captured by Frankle et al. (2020) – they show that instability to SGD noise i.e. the performance gap between networks A and B and their interpolants at convergence gradually decreases as $k$ increases. When $k$ becomes sufficiently large, the interpolated weights perform better than or approximately as well as networks A and B at all stages of training, making the network "stable to SGD noise". Intuitively, an increase in $k$ may correspond to gradually determining the "basin fate" of the network (Fort et al., 2020), explaining the decrease in the rate at which performance degrades. However, we believe that the picture is still far from clear and does not explain observation (a) at all.

One can also observe that in the later stages of training, interpolants can do just as well but do not outperform networks A and B – we address an aspect of this in Section 4.

## 4    Training Linear Interpolants

From a practical perspective, our findings so far lead to a natural question: if linearly interpolating the weights of networks trained on different sets of SGD noise leads to better performance (given appropriate values of $k$ and $s$), can we efficiently leverage this to speed up optimization?

We address this by looking at the simplest case – two copies of a network are made after $k$ epochs, trained on different sets of SGD noise, and then interpolated to produce a network with better performance. The resulting interpolant is then trained further to convergence. From the previous section, one may observe that the degradation or improvement in performance is most significant for $\alpha = 0.5$ – we will use this setting in our further experiments, though our observations generally hold for other reasonable values of $\alpha$ (i.e. not very close to 0 or 1).

We train ResNet-20 (He et al., 2016) networks on CIFAR-10 (Krizhevsky, 2009) with different values of $k$ and $s$ and evaluate their test performance with a focus on how performance changes at and after averaging network weights (see Fig. 3 and Fig. 9). In general, we find that while interpolating weights improves performance, it stagnates or degrades immediately over the next few epochs of further training. The subfigure corresponding to $k = 1$ and $s = 1$ is an exception to this rule but is not useful in pratice, since gains in performance very early in training are usually "easy" to achieve.

Simply dropping the learning rate after interpolation removes this observed stagnation/drop in performance, but negatively impacts the performance at convergence, which may be undesirable (see Fig. 12). As a compromise, we try warming up the learning rate after interpolation instead and find that this does not cause any noticeable difference in the originally observed behavior – performance continues to degrade when the learning rate is dropped and gradually increased. These results can be observed in Fig. 4.

These results suggest that networks obtained by weight interpolation consistently end up in regions of the loss landscape that are locally "better", but are not naturally amenable to further training through SGD – the degradation in the improved performance heavily implies that training is stalled as the interpolated network escapes its present position in the loss landscape. Furthermore, this phenomenon does not seem to be specific to a particular model architecture or choice of hyperparameters (such as the learning rate schedule), hinting that what we observe is something more fundamental to the optimization process.

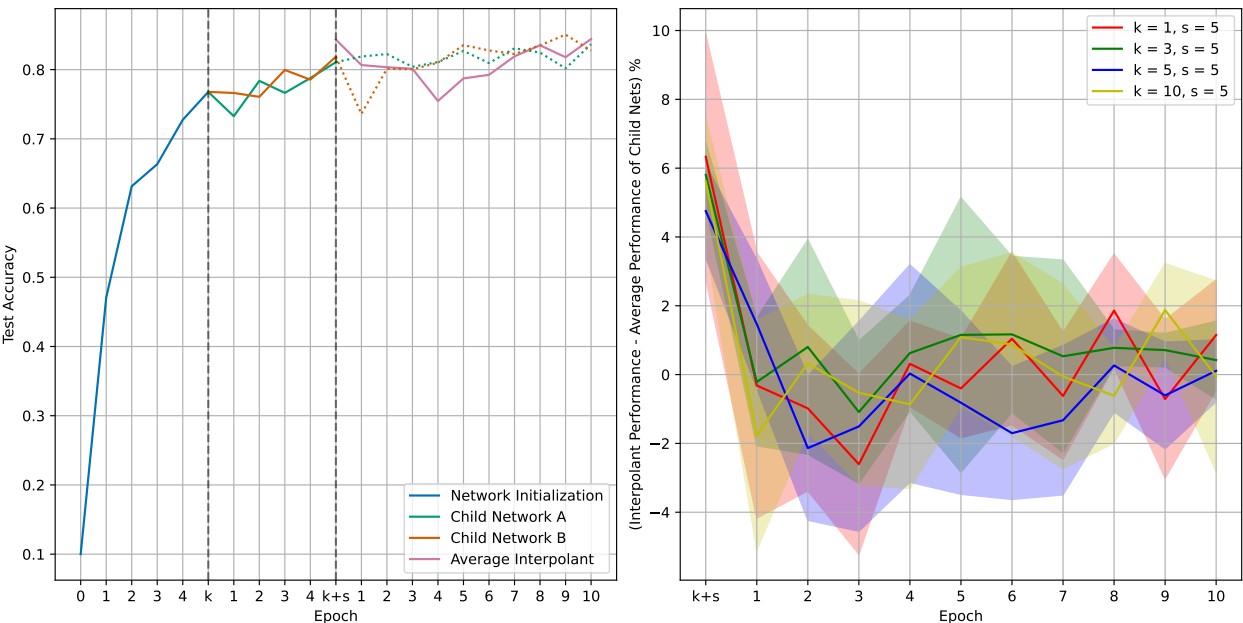

Figure 3: (a) Zoomed-in view of test performance of child networks and interpolant before and after weight averaging, for $k = 5$ and $s = 5$ epochs. We observe a rise in test accuracy followed by an immediate drop over the next few epochs. In addition to training the interpolant, we also train the child networks independently as a reference. (b) Difference between interpolant and child network performance, for various (sufficiently large) values of $k$. We consistently observe that the interpolant performs significantly better than the child networks at the point of interpolation, but there is effectively no difference after 1-2 epochs. See Fig. 9 for variations.

Our results so far raise **two key questions**:

- Given sufficiently large values for $k$ and $s$, why does linear interpolation of network weights trained from the same initialization on different sets of SGD noise lead to networks that perform consistently and significantly better? Answering this can potentially lead to finding an efficient heuristic/method to ascertain appropriate values for $k$ and $s$.

- Why are interpolants derived in this manner not naturally amenable to further training through SGD? What are the properties of interpolants and their immediate positions in weight space or the loss landscape that cause this, and if and how can this be prevented or mitigated?

## 5 Example Importance

We now consider the following questions: Are there specific examples in the training set that are impacted more heavily than others by weight interpolation? If so, is it the same set of examples that contribute significantly to the improvement and subsequent drop in performance?

Frankle et al. (2020) note that networks become stable to SGD noise early but at different points in training for different datasets during the training process. Furthermore, Paul et al. (2021) introduce metrics (see Section 5.1) to measure example importance for training and achieving good generalization. The authors show that data subsets differentiated by these metrics become stable to SGD noise at significantly different points in training – in general, the more important the example, the later in training stability is achieved with respect to it. Motivated by this, we explore how performance changes at and after interpolation on subsets of data of similar importance according to this metric.

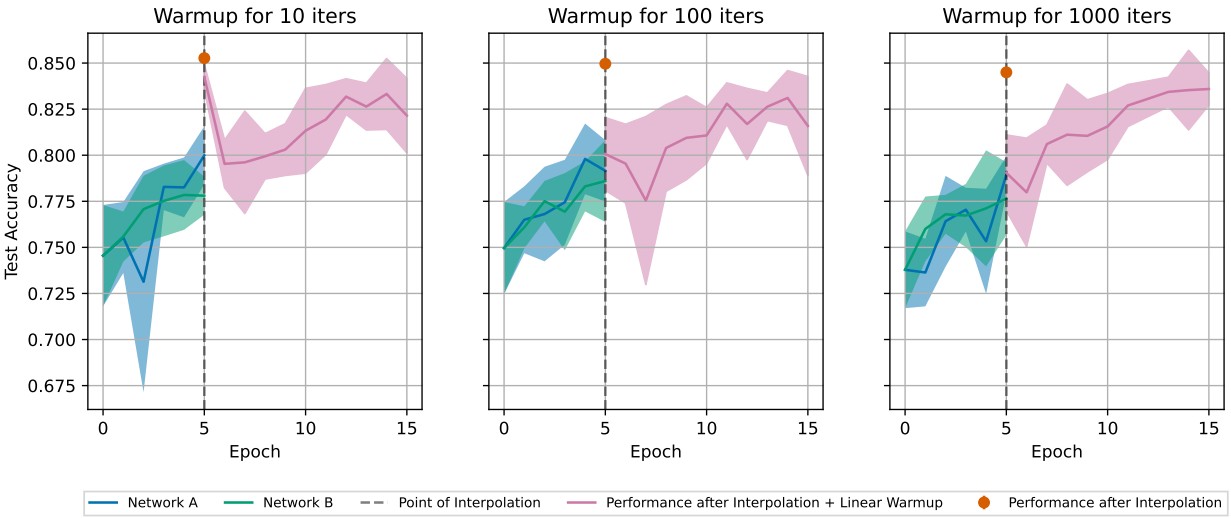

Figure 4: Zoomed-in view of test performance of child networks before interpolation and the weight-averaged network at, and after interpolation, for a variety of $k = 5$ and $s = 5$ epochs – note that the plot begins *after* the first $k$ epochs. Test performance of the averaged network continues dropping even when linear warmup is employed after interpolation. The learning rate is warmed up over $t$ iterations, where the learning rate is $\frac{i}{t} \times l$ at iteration $i$, and $l$ is the original learning rate.

## 5.1 EL2N Score

Introduced by Paul et al. (2021), the Error L2-Norm (EL2N) score of a training example is defined as the expected norm of the difference between its label as predicted by the model and its ground truth label, where the expectation is computed at some early epoch $t$ over multiple trained network initializations. The authors demonstrate that the higher-scoring examples are *more difficult* to learn but are also more *important for generalization* – removing these difficult examples from the training set had the biggest effect on the final generalization of the model.

## 5.2 Interpolant Performance Improves and Degrades on Important Examples

We maintain the experiment setup from the previous section, where child networks (ResNet20) are trained on different sets of SGD noise, averaged, and then trained further. In Fig. 5, we track the performance of the child networks and the averaged network across 4 equal splits of CIFAR-10's training set. The data is split according to the EL2N scores of the training examples computed at epoch 10 of training – each split consists of 12500 examples, with Split 1 and Split 4 containing examples with the lowest and highest EL2N scores respectively.

As expected and shown by previous work, more important examples (as defined above) are indeed learned later in training. We also observe that it is primarily on Splits 3 and 4 that the network's performance improves and subsequently drops. While the former may seem obvious (these are the only splits where there is space for performance to improve), the latter is not – linearly interpolating the weights causes a drop in performance almost exclusively on the harder examples in a dataset. Furthermore, the network maintains stellar performance on the easier examples when trained further.

Our observations emphasize the connection between interpolated network performance and stability to SGD noise on subpopulations of data differentiated by example importance – interpolants derived from networks trained on separate SGD noise trajectories perform better on data subpopulations that are still unstable to SGD noise. On the other hand, once stability to SGD noise is reached on a specific subpopulation, there is no degradation in performance from weight interpolation.

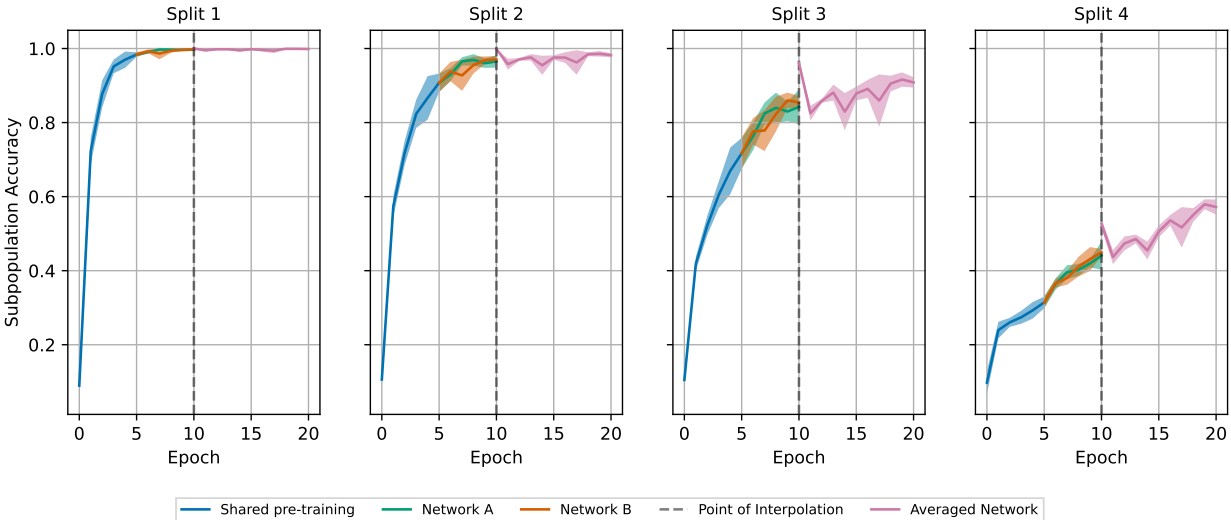

Figure 5: Performance of a ResNet20 network on 4 splits of CIFAR-10's training set differentiated by EL2N score, where Split 1 contains the least important examples, and Split 4 contains the most important ones. The rise and subsequent drop in performance due to interpolation is most noticeable in Split 3 and Split 4, which the network performs relatively poorly on. As noted in prior work (Paul et al., 2021), the network is likely to be unstable to SGD noise with respect to Splits 3 and 4 at such an early point in training, providing more evidence for the connection between our observations and stability to SGD noise.

## 6 Conclusion and Discussion

We have empirically demonstrated that when copies of a neural network are trained on different sets of SGD noise, the network resulting from a linear interpolation of their weights can perform better than the trained copies. We frame this observation as an extension of linear mode connectivity, offering a more complete picture of how networks become stable to SGD noise. Interestingly, when such an interpolated network is further trained, performance fails to improve or degrades over the next few epochs. We also demonstrate that strategies such as dropping or warming up the learning rate do not remedy this observation – linearly interpolated networks consistently end up in brittle areas of the loss landscape that make the network perform better in the short term but impede further optimization. We analyze this phenomenon from the lens of example importance and empirically demonstrate that the observed improvement and degradation in performance occur specifically in data subsets that are more important for generalization, providing further evidence for the connection between our observations and differences in stability to SGD noise on examples differentiated by example importance.

We conclude by noting some limitations and directions for future work. The biggest question raised by our work is: Why does a simple linear interpolation of network weights trained on different sets of SGD noise consistently show better performance, and how can we effectively leverage this improvement? While we connect our observations to prior work, we do not comprehensively answer this question. The strategies we explore to mitigate the drop in performance when training the interpolants are based on previously used optimization techniques and tricks; due to the empirical nature of our study, we cannot conclude whether an alternative optimization method exists that could robustly optimize the interpolants, exploiting the drop in error to speed up training. It is also important to note that the current observations only apply to the standard vision tasks, and it is an empirical question whether they would extend to other domains.

Overall, our work provides insight into the behavior of network weight interpolations during the early stages of training and at the same time, raises new questions to address. We believe that this will help us better understand neural network loss landscapes and weight interpolation in the future.

**Acknowledgments**

This research was supported by an NSERC Discovery grant and the Canada CIFAR AI Chairs program. The authors acknowledge material support from NVIDIA and Intel in the form of computational resources and are grateful for technical support from the Mila IDT team in maintaining the Mila Compute Cluster. The authors would also like to thank Devin Kwok for helpful feedback and discussions in the early stages of this work, and Amr Khalifa for feedback on a draft.

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

## A  Experimental Setup

For all experiments, networks were trained with a batch size of 128 for a total of 160 epochs. A stepwise learning rate schedule was employed with an initial learning rate of 0.1 – the learning rate is reduced by a factor of 10 at epoch 80 and epoch 120. When linear warmup is employed over $t$ iterations, the learning rate is $\frac{i}{t} \times l$ at iteration $i$ where $l$ is the initial learning rate. All datasets were also augmented with random crops to a size of 32x32 pixels after padding 4 pixels to each border in the original image, with a constant value of 0, and horizontal image flips with a probability of 0.5.

## B  Results using LayerNorm

Relative to BatchNorm, LayerNorm does not enable large learning rates – as a consequence, we cannot make use of the same training setup. In order to draw a fair comparison between the two cases, we linearly warm up the learning rate to 0.1 over 100 iterations. We attempt to keep the number of warmup iterations as small as possible since warmup is known to induce linear mode connectivity onset more quickly Frankle & Carbin (2019); Frankle et al. (2020). We note that the performance gain due to interpolation and drop after interpolation is smaller compared to the BatchNorm case – we leave a more complete analysis of this for future work.

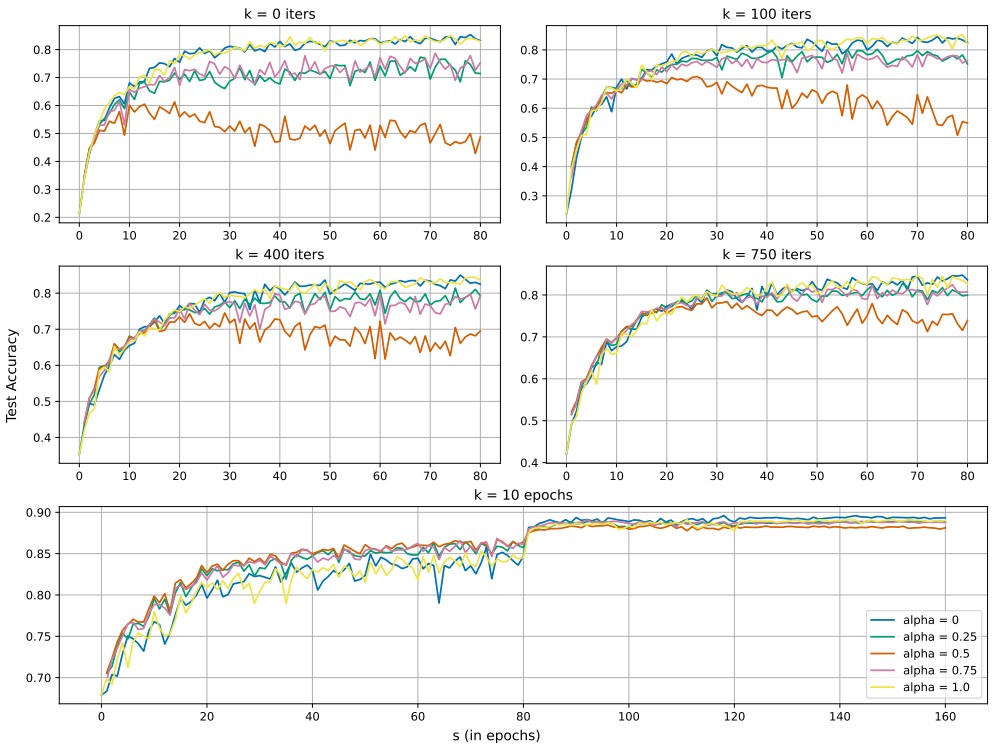

Figure 6: Progression of test performance of networks resulting from linear weight interpolation of child networks.

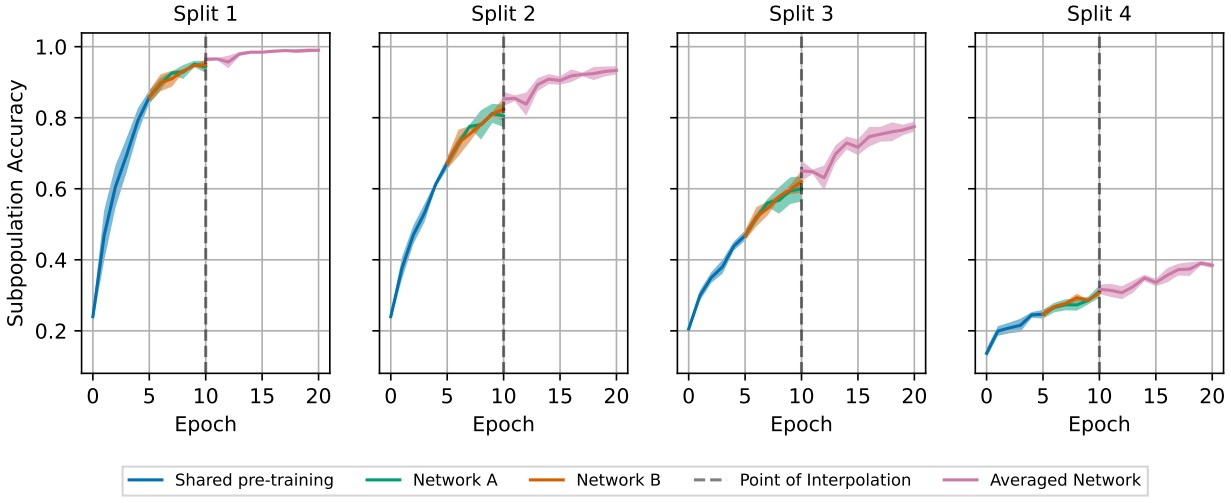

Figure 7: Performance of a ResNet20 network on 4 splits of CIFAR-10's training set differentiated by EL2N score.

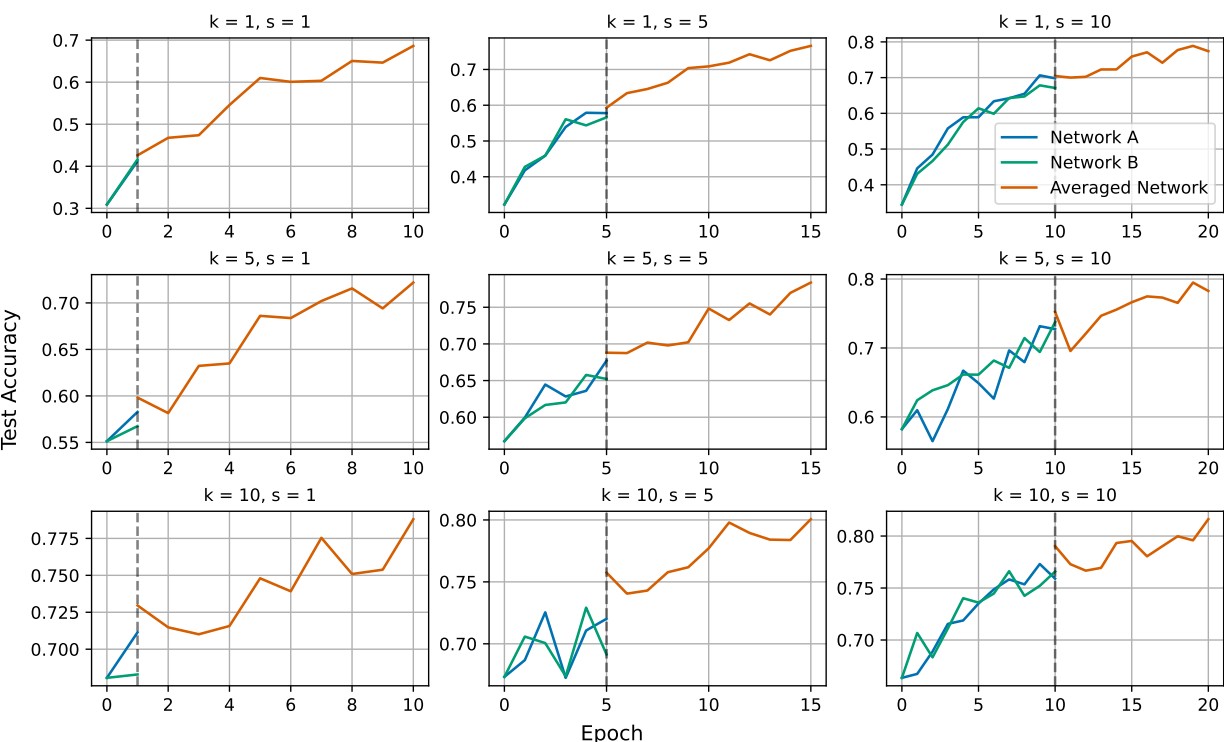

Figure 8: Test performance of child networks before interpolation and the weight-averaged network at, and after interpolation, for a variety of k and s values in epochs.

## C   Relative Change in Accuracy and Error on Data Splits

| Split | By Interpolation | | After Interpolation | |
|---|---|---|---|---|
| | $\Delta$ Accuracy | $\Delta$ Error | $\Delta$ Accuracy | $\Delta$ Error |
| Split 1 | 0.00285 | -0.98367 | -0.0049 | inf |
| Split 2 | 0.03166 | -0.94480 | -0.04067 | 27.21314 |
| Split 3 | 0.13364 | -0.7386 | -0.14097 | 3.6074 |
| Split 4 | 0.18621 | -0.14679 | -0.17065 | 0.19244 |

Table 1: Relative change (denoted by $\Delta$) in accuracy and error due to weight interpolation and just after weight interpolation, on data splits based on difficulty. The former is computed using the difference between interpolant accuracy/error and the mean accuracy/error of the child networks. The latter is computed using the difference between interpolant accuracy/error just after interpolation, and after training it for one epoch. The results are averaged over 5 runs. Note that $\Delta$ Error After Interpolation for Split 1 is reported as inf since the interpolant achieves perfect accuracy (and consequently zero error) before being trained in nearly all runs considered.

## D   Additional Results for ResNet20 + CIFAR-10

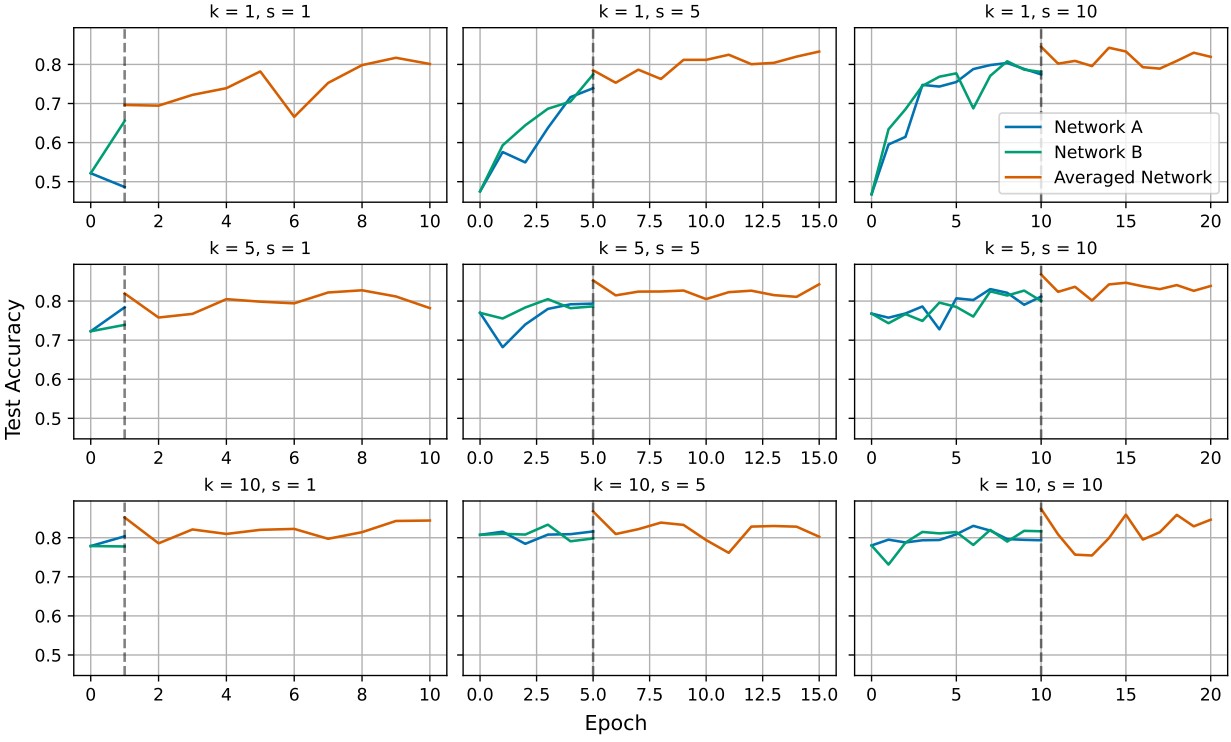

Figure 9: Zoomed-in view of test performance of child networks before interpolation and the weight-averaged network at, and after interpolation, for a variety of $k$ and $s$ values in epochs. We consistently observe a rise in test accuracy followed by an immediate pause in improvement or drop over the next few epochs. Interestingly, the magnitude of improvement seems to be independent of the exact values of $k$ and $s$.

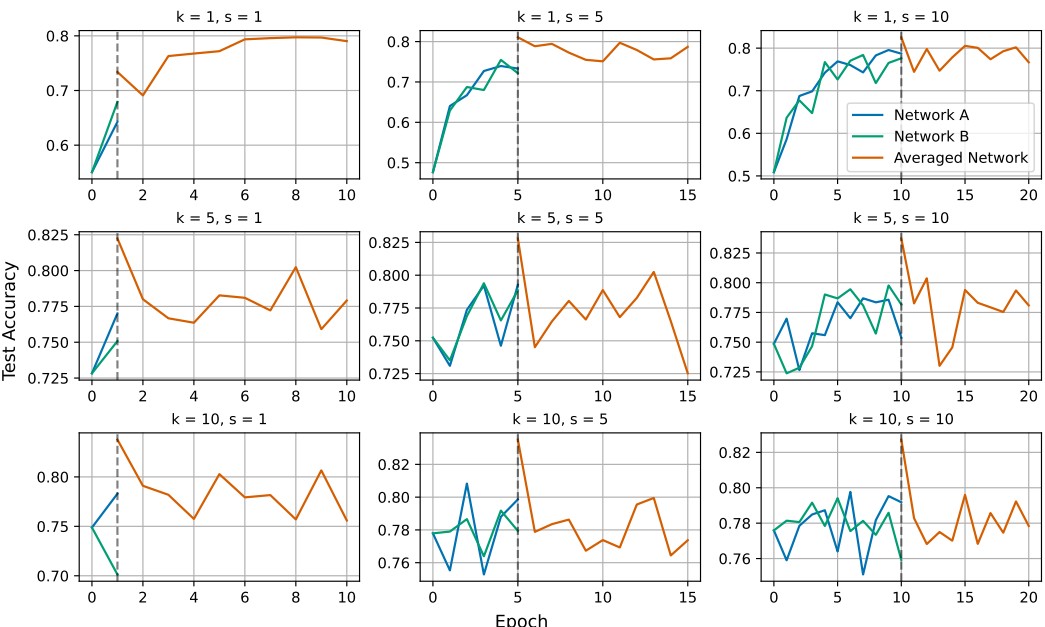

Figure 10: The setup remains the same as in Figure 9, except no augmentation is used i.e. SGD noise consists of only the randomness due to data orders. Our observations are still characterized by a rise and subsequent fall in performance – removing augmentations from the training setup only makes this behavior more noticeable.

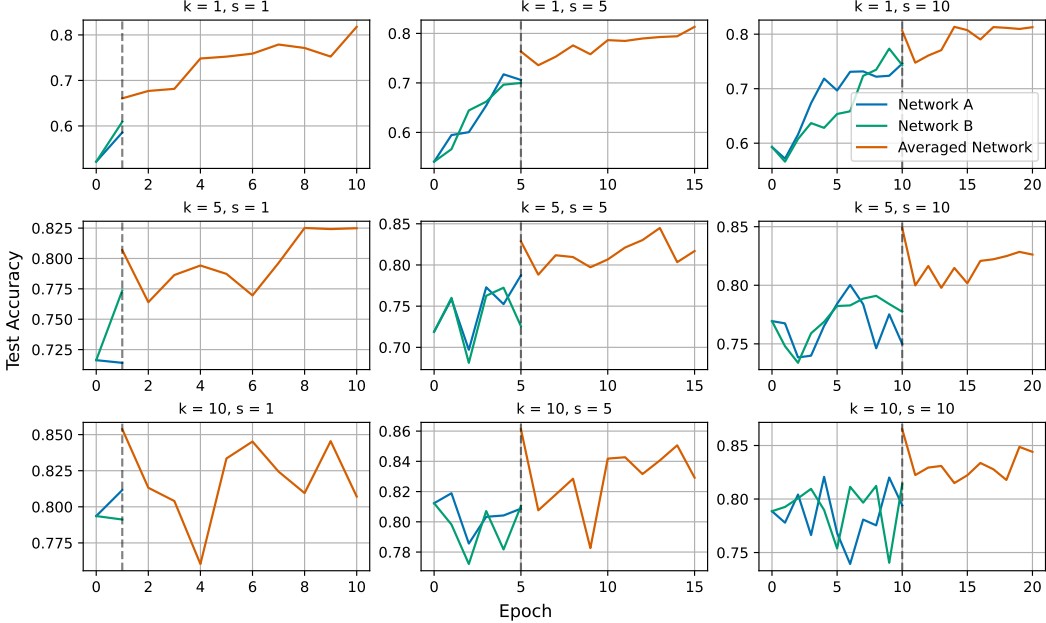

Figure 11: The setup remains the same as in Figure 9, except child networks are trained on non-overlapping, equally sized random splits of the training data. To be specific, the initialized network is trained on the entire train set in the $k$ phase, the child networks are trained on splits in the $s$ phase, and the resulting interpolant is trained on the entire dataset. Once again, our observations largely remain the same.

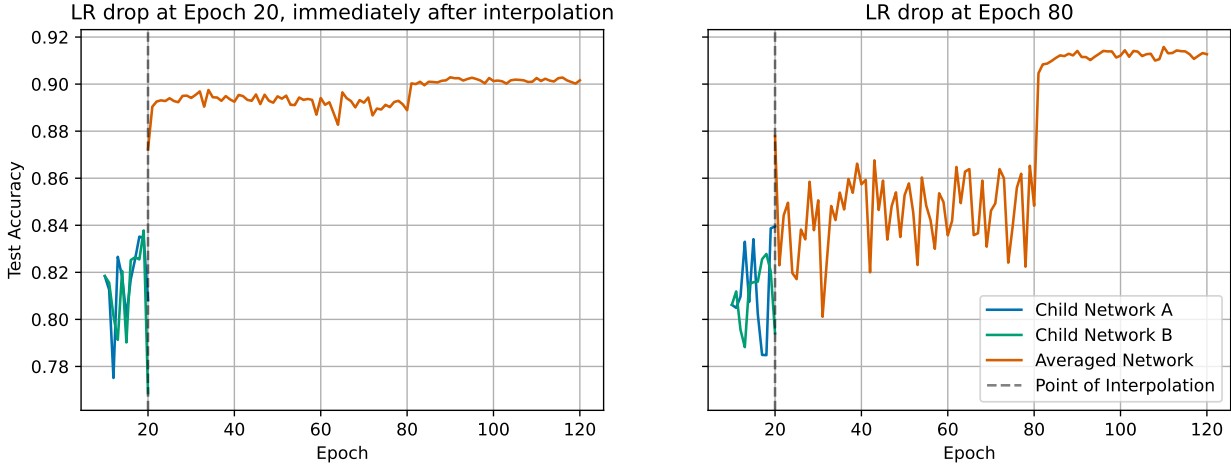

Figure 12: Dropping learning rate early in training removes the performance degradation observed in Fig. 3 but affects final performance significantly.

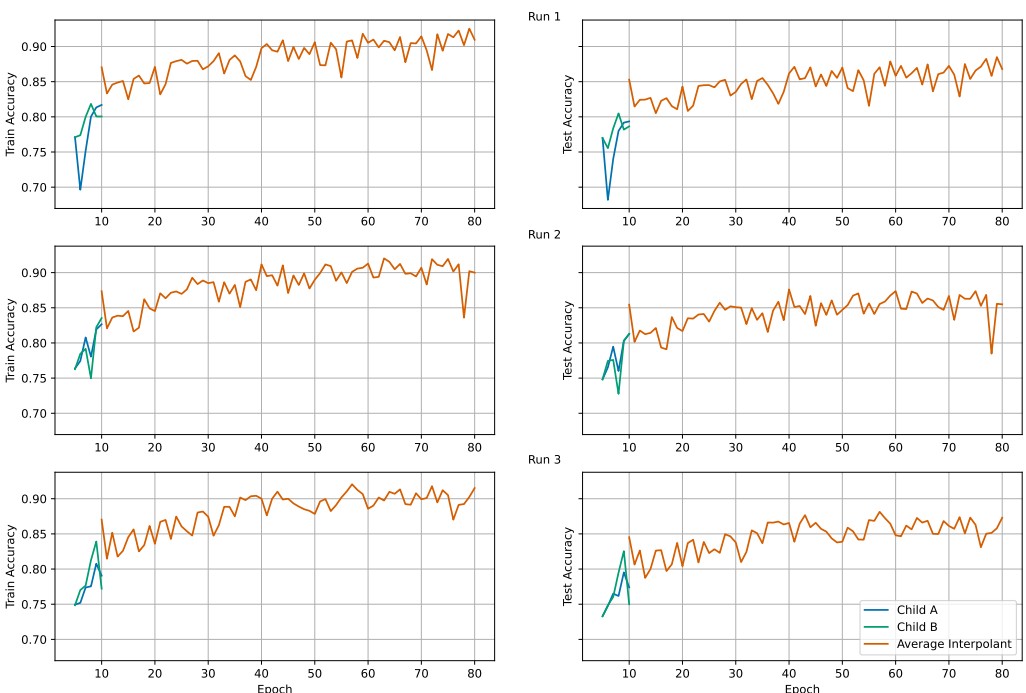

Figure 13: Extended training runs of interpolants with a constant learning rate – note that training accuracy non-monotonically yet consistently increases as training progresses, implying that the drop in performance after interpolation is not due to overfitting.

## E    VGG-16 + CIFAR-10

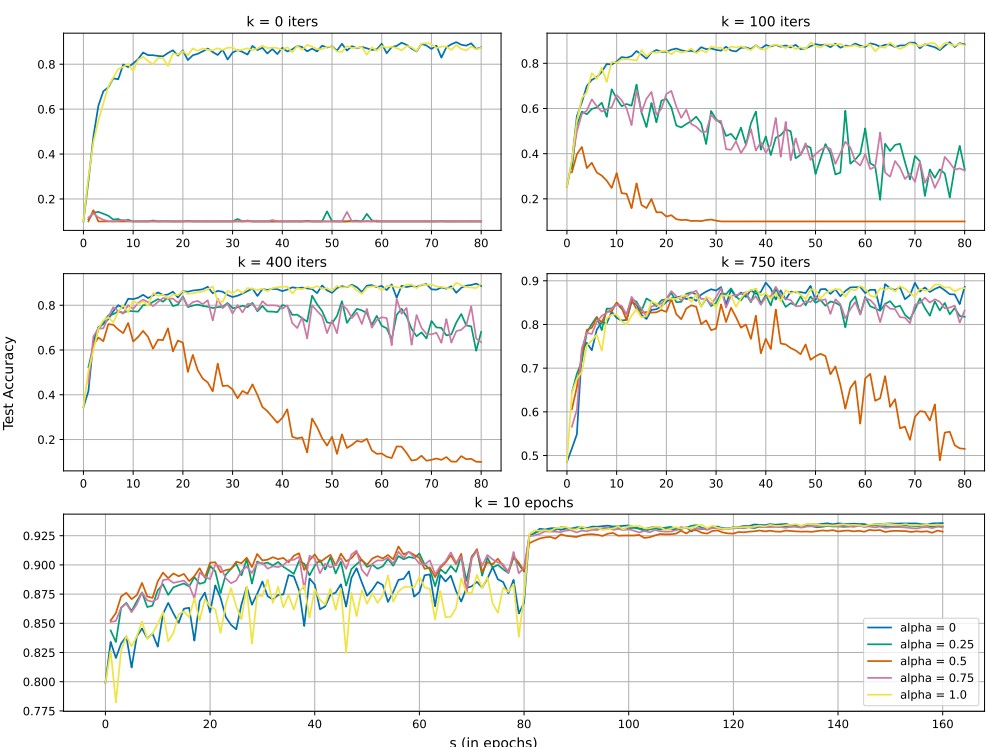

Figure 14: Progression of test performance of networks resulting from linear weight interpolation of child networks.

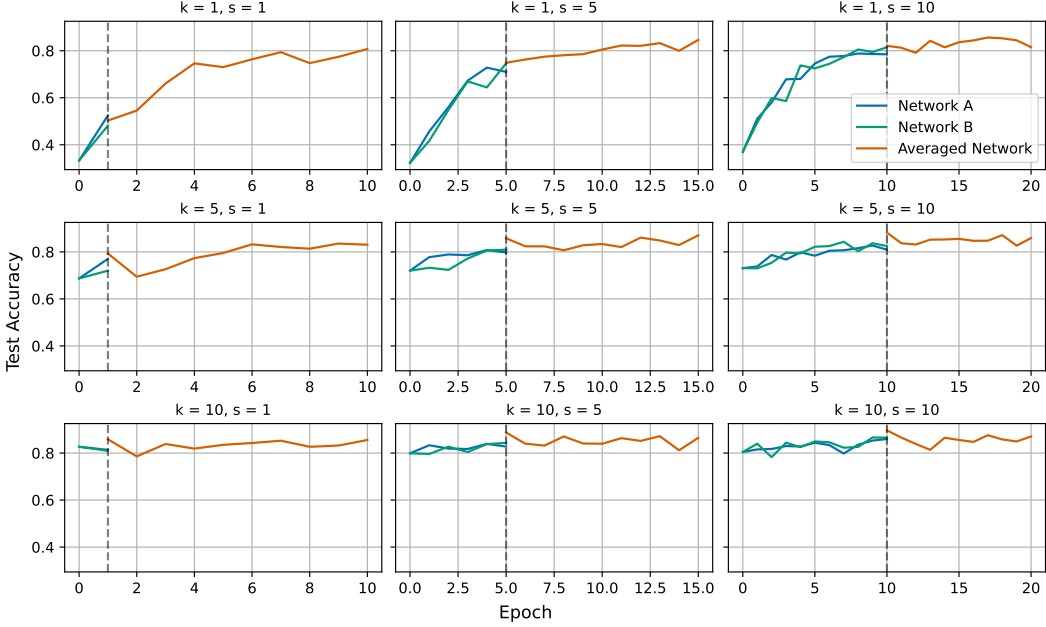

Figure 15: Test performance of child networks before interpolation and the weight-averaged network at, and after interpolation, for a variety of k and s values in epochs.

## F  ResNet56 + CIFAR-100

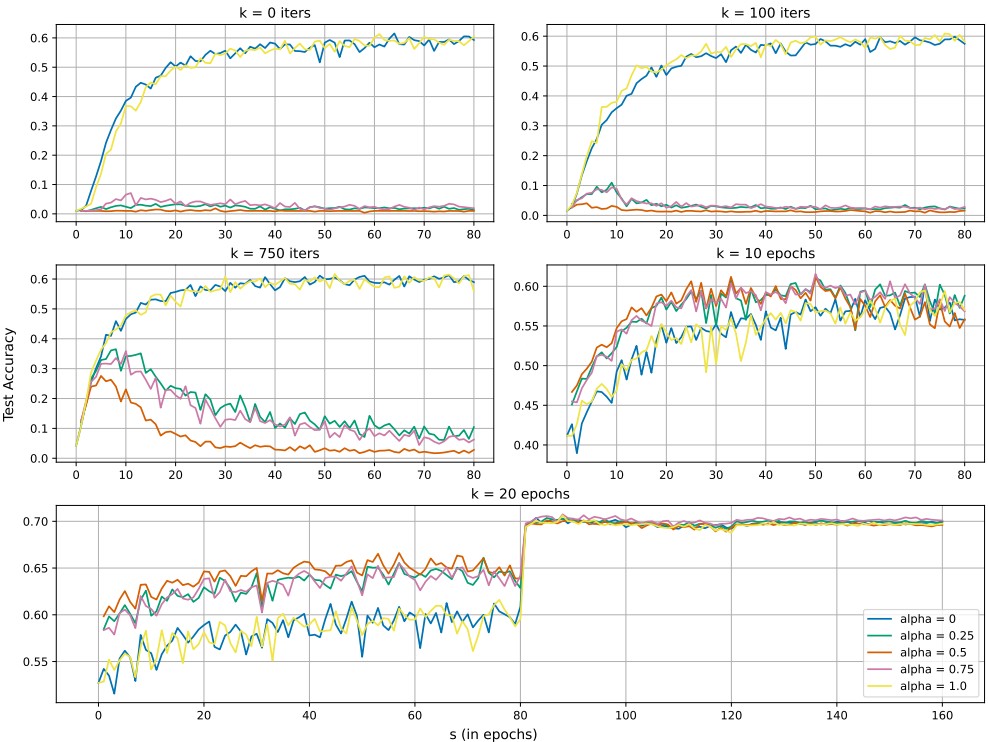

Figure 16: Progression of test performance of networks resulting from linear weight interpolation of child networks.

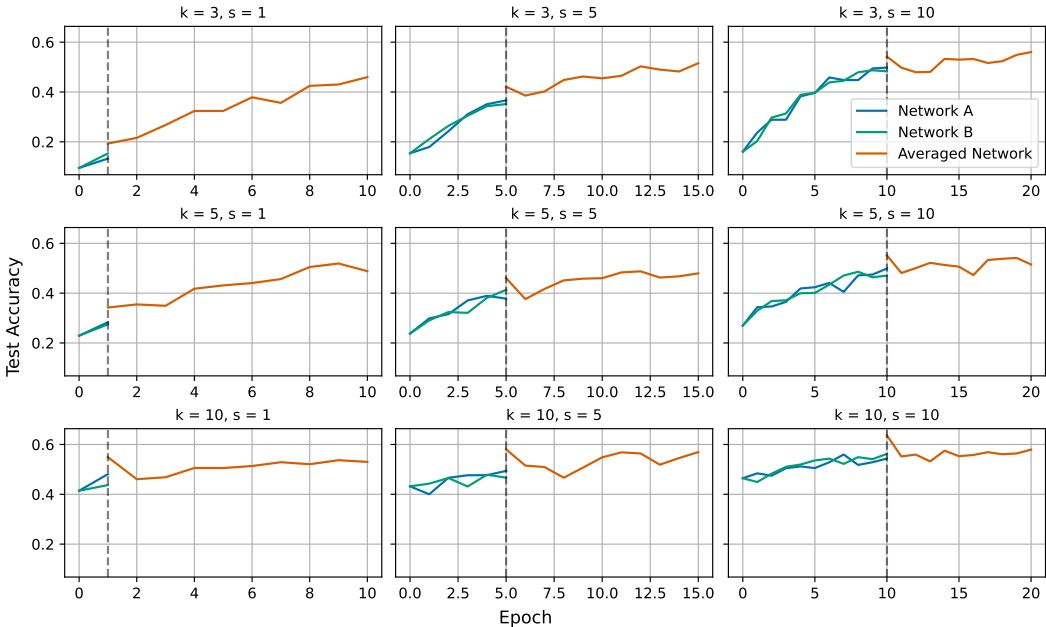

Figure 17: Test performance of child networks before interpolation and the weight-averaged network at, and after interpolation, for a variety of k and s values in epochs.

## G   ResNet20 + CINIC-10

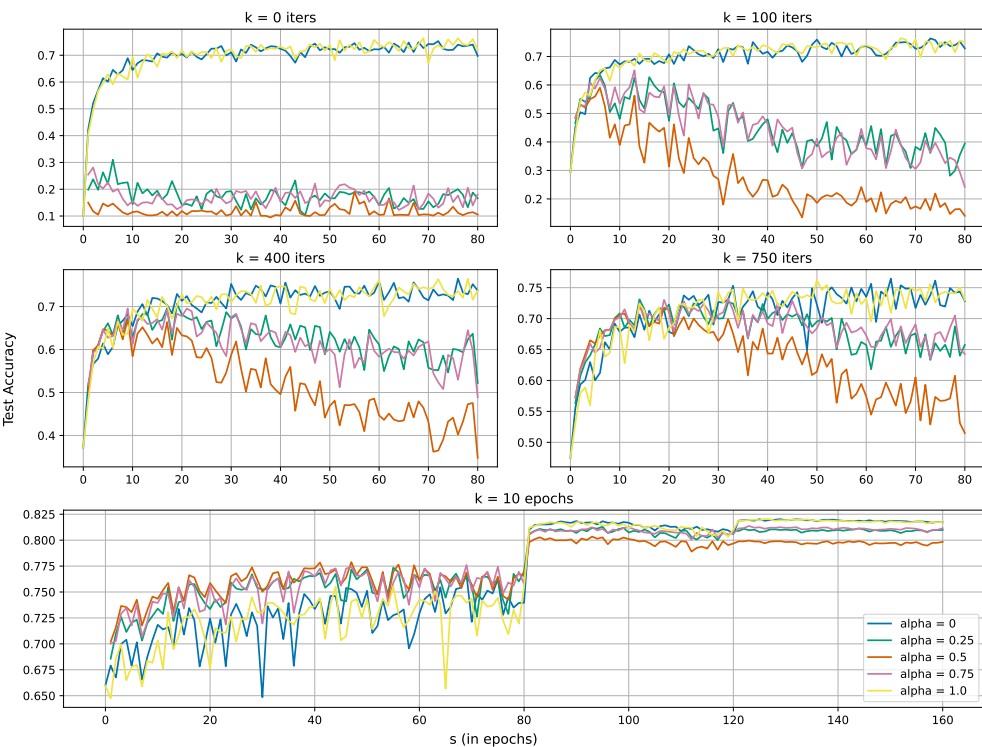

Figure 18: Progression of test performance of networks resulting from linear weight interpolation of child networks on CINIC-10 (Darlow et al., 2018) with a ResNet20.

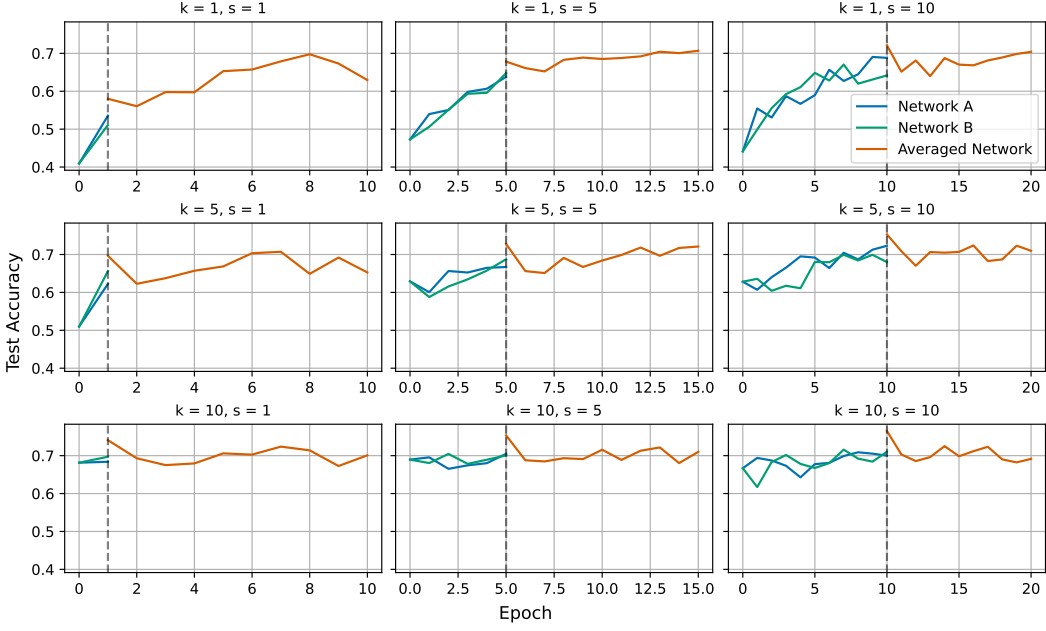

Figure 19: Test performance of child networks before interpolation and the weight-averaged network at, and after interpolation, for a variety of k and s values in epochs on CINIC-10 with a ResNet20.

