# OpenReview forum: "Linear Weight Interpolation Leads to Transient Performance Gains"
_TMLR — Accepted by TMLR_

### Review · Reviewer_8TwW · 2024-07-13

**Summary Of Contributions:**

This work empirically shows that that linear interpolation leads to a performance gain immediately after merging, and this gain disappears if one finetunes the merged model

**Audience:**

Yes

**Claims And Evidence:**

No

**Requested Changes:**

See the weakness section

**Strengths And Weaknesses:**

Strength: The result is clean and well-explained. The observed phenomenon is novel and can have interesting empirical and theoretical implications

Weakness: I feel that the result and its analysis are a little too shallow and straight forward and can be made much deeper. For example, the authors should (1) conduct experiments to show why there is a gain and why there is a drop? To me (and perhaps most of the audience), it appears that the drop is due to simple overfitting, is this true or not? The authors should offer an answer empirically

Besides, the gain on split 3 is much more significant on split 4. There should also be an explanation

Lastly, I think comparing in the raw accuracies is misleading. The authors should report the **relative** change in the **prediction error**, which makes it possible and reasonable to compare between split 1 and other splits. It might well be the case that split 1 is more affected in this metric

---

> ### Author Response · Authors · 2024-08-13
> **Response to Reviewer 8TwW**
>
> We thank the reviewer for their very helpful feedback. We individually address comments below:
>
> > I feel that the result and its analysis are a little too shallow and straightforward and can be made much deeper. For example, the authors should (1) conduct experiments to show why there is a gain and why there is a drop?
>
> Thank you for making this suggestion. We agree that it would be interesting and instructive to work out why exactly we observe the gain and drop in performance. However, our experiments thus far have suggested that this question may be quite difficult to answer, and we have therefore concentrated in the present paper on thoroughly documenting the observed behavior. As a small step, we have added results of experiments using LayerNorm instead of BatchNorm in Appendix B -- we observe that the performance gain and drop are noticeably smaller in this setting, though all the same trends occur. While this certainly does not solve the puzzle, we hope that this, along with our experiments on data difficulty, will offer us and the broader community promising directions to explore in future work.
>
> > To me (and perhaps most of the audience), it appears that the drop is due to simple overfitting, is this true or not? The authors should offer an answer empirically.
>
> Good question! In Appendix D, we add plots showing interpolant training performance over a large number of epochs with a constant learning rate, over 3 different training runs -- one can observe that the drop due to interpolation occurs in the training accuracy (not just the validation accuracy) and afterward continues to increase steadily, implying that the effect is not caused by the networks overfitting to the dataset.
>
> > Besides, the gain on split 3 is much more significant on split 4. There should also be an explanation.
>
> > Lastly, I think comparing in the raw accuracies is misleading. The authors should report the relative change in the prediction error, which makes it possible and reasonable to compare between split 1 and other splits. It might well be the case that split 1 is more affected in this metric.
>
> This is an excellent observation and suggestion. We will address the above two points together:
>
> Firstly, in Appendix C, we report relative changes in accuracy and error both at and immediately after the point of interpolation. We believe it is important to provide both these metrics since they follow opposing trends (in terms of magnitude) – Split 1 sees the lowest relative change in accuracy, but largest in terms of error and vice-versa for Split 4. Therefore, in one sense Split 4 is indeed impacted the most, and in another, the least.
>
> Secondly, we speculate why Split 4 might be less impacted than Split 3 in terms of raw accuracies. [Paul et al.](https://openreview.net/pdf?id=Uj7pF-D-YvT) note the following:
> ```
> ...the very highest scoring examples (that) might be too difficult to learn, perhaps because they are unrepresentative samples...
> ```
> Indeed it is likely that these examples belong to Split 4. If weight interpolation learns (and unlearns) something that improves generalization, the unrepresentative samples would possibly be less impacted by it. We re-emphasize that this is speculative, and it would be a worthwhile future direction to rigorously ascertain the reason.

---

### Review · Reviewer_uKDG · 2024-07-22

**Summary Of Contributions:**

The authors consider vision tasks and analyse the performance of neural networks whose weights are obtained by interpolating the weights of two trained networks. Specifically, the authors train a network for a small number of SGD iterations $k$, after which two copies of the network are trained independently on the same data set with different realisations of SGD noise (i.e. data order and data augmentations) for $s$ epochs. This work thus builds on the earlier work by Frankle et al. (ICML 2020) who studied interpolants at convergence.

Here, the authors show that early in training, interpolated networks outperform the network from which the interpolants were cloned from, but deteriorates later in training. Interestingly, they show that the both the initial boost in performance and the later deterioration depend on the same examples. They identify these examples using the Error L2-Norm score (EL2N) introduced by Paul et al. (NeurIPS '21). The EL2N score is the expected difference in $\ell_2$ norm between model prediction and input label averaged over multiple networks after training for a few epochs $t$. Here, the authors show that interpolant performance improves and degrades on inputs with large EL2N score.

**Audience:**

Yes

**Broader Impact Concerns:**

I do not think the work raises ethical implications that should be addressed separately.

**Claims And Evidence:**

Yes

**Requested Changes:**

I think the manuscript is ready for publication in its current form.

**Strengths And Weaknesses:**

- The authors present interesting results pertaining to model averaging and the stability of trained models to SGD noise, both topics which have received significant interest in the community served by TMLR.
- The experimental results are clean and convincing, and support the main results of the paper. The authors also perform several ablations on their results, investigating the impact of learning rate schedules.
- The writing of the article is clear.
- The authors put their work in the context of recent work on interpolating networks, and on example difficulty etc (although I should note that I'm not an expert in this literature!)

---

> ### Author Response · Authors · 2024-08-13
> **Response to Reviewer uKDG**
>
> We thank the reviewer for their comments and feedback, and are glad that the manuscript is compelling in its current form.

---

### Review · Reviewer_CxPM · 2024-07-30

**Summary Of Contributions:**

This paper investigates the behavior of *interpolant* neural networks, which are defined as networks derived by linearly interpolating the weights from two constituent networks A and B. The interpolant network may then continue to be trained as a single network, or its constituent networks can continue to be trained separately.

The authors produce a collection of empirical findings, including that interpolant networks can outperform their constituent networks but that this performance can degrade with subsequent additional training after the point of interpolation. This behavior appears to be sensitive both to the timing of when interpolation occurs and to how much additional training takes place, and it varies depending on how "hard" samples are to learn. The authors relate these observations to previously described phenomena of linear mode connectivity and stability to SGD noise.

As a caveat to this review, I have no expertise in neural network interpolation and am unfamiliar with all but one of the works cited.

**Audience:**

Yes

**Broader Impact Concerns:**

None.

**Claims And Evidence:**

Yes

**Requested Changes:**

- Some clarification of the above point, either in the paper or in the comments (if I've possibly misunderstood something).
- Figure 2 caption: $\alpha$ should be the interpolation weight, not the timestep.
- End of section 3: observations (a) and (b) aren't actually labeled as such.

**Strengths And Weaknesses:**

This is an extremely well written and cohesive paper. Despite my lack of background knowledge in this area, I feel after reading it that I understand the problem, its significance, and at least some of the unresolved questions in this area.

Taking the authors at their word about what work has been done previously, it appears that they are the first to investigate "early" interpolation of networks, including with the two post-interpolation training variants: continuing to train the constituent networks or training the interpolated network itself. Their findings are non-obvious (to me at least) and raise interesting questions about where interpolant networks lie in the loss landscape relative to their constituent networks.

While this is an empirical paper with no theoretical results, the authors do a very good job of relating their findings to previously observed phenomena, offering compelling (if necessarily speculative) explanations of the interpolant networks' behavior.

The experiments are clear and appropriate for investigating the questions of interest, and the findings are certainly of interest for understanding neural network loss landscapes and convergence behavior.

My one quibble is that I'm slightly confused by the first "key question" on page 6: "Why does linear interpolation of network weights trained from the same initialization on different sets of SGD noise lead to networks that perform consistently and significantly better? How does this depend on quantities $k$ and $s$?" Looking at Figure 2, it seems that if $k$ isn't large enough, then the interpolant networks only perform better for a very small number of additional epochs $s$, and their performance continues to degrade in $s$. It seems like this claim, which is repeated in several places, needs to be appropriately qualified or at least clarified. I think the authors do this elsewhere (e.g. "given appropriate values of $k$ and $s$"), but it would be easy for a reader to lose track of this caveat.

---

> ### Author Response · Authors · 2024-08-13
> **Response to Reviewer CxPM**
>
> We thank the reviewer for their very helpful feedback. We individually address comments below:
>
> > My one quibble is that I'm slightly confused by the first "key question" on page 6: "Why does linear interpolation of network weights trained from the same initialization on different sets of SGD noise lead to networks that perform consistently and significantly better? How does this depend on quantities $k$ and $s$?" Looking at Figure 2, it seems that if $k$ isn't large enough, then the interpolant networks only perform better for a very small number of additional epochs $s$, and their performance continues to degrade in $s$. It seems like this claim, which is repeated in several places, needs to be appropriately qualified or at least clarified. I think the authors do this elsewhere (e.g. "given appropriate values of $k$ and $s$"), but it would be easy for a reader to lose track of this caveat.
>
> This is a good point. The reviewer is correct in noting that our original statement should make explicit the role of $k$ and $s$.
>
> We have rewritten this key question, which we hope will provide sufficient clarity (see below):
>
> ```
> Given sufficiently large values for k and s, why does linear interpolation of network weights trained from the same initialization on different sets of SGD noise lead to networks that perform consistently and significantly better? Answering this can potentially lead to finding an efficient heuristic or method to ascertain suitable values for k and s.
> ```
>
> > Figure 2 caption: $\alpha$ should be the interpolation weight, not the timestep.
>
> We have made this change -- thank you for the suggestion!
>
> > End of section 3: observations (a) and (b) aren't actually labeled as such.
>
> We have made the labeling more explicit in the revised paper -- thank you!

---

### Author Response · Authors · 2024-09-27
**Camera-Ready Revision**

We would once again like to thank the action editor and reviewers for their valuable and constructive feedback!

---

### Decision · Action_Editor_LdBD · 2024-08-24

**Recommendation:** Accept with minor revision

**Comment:**

Reviewers were anonymous in recommending this paper for acceptance. Since this paper is on a topic that is near and dear to me, I gave it a full read too. I think it provides a very simple additional insight into what I think may be an important phenomenon - namely, that SGD doesn't necessarily find the best solution (in terms of either train or test loss) and that a very simple operation (averaging weights) can find better solutions. But interestingly, and "novely", this paper shoes that such gains go away if you train with SGD some more. The paper summarizes it best: "These results suggest that networks obtained by weight interpolation consistently end up in regions of the loss landscape that are locally “better”, but are not naturally amenable to further training through SGD". This is an interesting result, if a bit of a bite-sized one. The paper includessSome additional analysis into which datapoints are impacted by this phenomenon, which I think are also interesting though somewhat unsurprising. The paper is written clearly and is easy to understand.

I think the only weakness of the paper is its lack of depth (noted by one reviewer). Most notably, the experiments are in what is now a relatively "toy"/old-school setting - training ResNets on CIFAR-10. While there are certainly some applications of ML that involve training similarly small models on similarly small datasets from scratch, the setup is a pretty far cry from current standard practice (which mainly involves fine-tuning much larger models). I think some of the results in the paper likely carry over to more realistic settings, but it does make me wonder how much of the results are specific to their setting. Heaviest on my mind is whether data augmentation has anything to do with it, because if the datapoints are differently stochastically augmented in each of the child networks after "splitting" training, then they actually aren't seeing the exact same data. Showing what happens when there's no augmentation and/or when the data is actually truly different would be interesting. The latter setting is most similar to e.g. federated learning/branch-train-merge, so would be especially interesting to know. I looked to the appendix to see how thorough/"heavy" the augmentation was, and it says "augmented with random
crops to a size of 32×32 pixels" - this is a little confusing because CIFAR-10 images are 32x32, I assume you mean you did some zooming first before cropping? Totally separately, it also makes me wonder optimizers that specifically attempt to influence the "sharpness" of the solutions would impact this (because, in a cartoon sense, SGD will only find a solution at the "edge" of a flat basin; interpolating will move into the middle of it; sharpness-aware minimization perhaps would also move to the middle of the basin)?

I don't necessarily think the paper needs to go into all of the questions I mentioned above (though I would ask that you clarify the specific kind of augmentation used). I think the paper would be considerably stronger even with a small amount of additional experiments (e.g. with no augmentation, or with completely different data) and would be quite a lot stronger if it considered more experimental settings (say, language models, or sharpness-aware minimization). I think these experiments are optional for the purpose of acceptance, but I nevertheless encourage the authors to consider them to amplify the paper's impact/reach.

**Audience:**

Yes, this paper provides some interesting results in an area that members of the TMLR community care about and that additionally may shed light onto some very important and understudied phenomena.

**Claims And Evidence:**

Reviewers all felt convinced by the results in this paper and considered the presentation of the results to be clear and convincing. I think there's a little more that the authors could have done to strengthen their claims (more on that in the "comment"), but there is no issue here.

---

> ### Author Response · Authors · 2024-09-27
> **Response to Action Editor LdBD**
>
> We thank the action editor for their very helpful feedback. We individually address comments below:
>
> > I looked to the appendix to see how thorough/"heavy" the augmentation was, and it says "augmented with random crops to a size of 32×32 pixels" - this is a little confusing because CIFAR-10 images are 32x32, I assume you mean you did some zooming first before cropping?
>
> Good catch! The augmentation is indeed a 0-padding of 4 pixels to image borders, followed by a random crop of 32x32 pixels. We have corrected this in the submitted camera-ready version as well -- thank you for pointing this out.
>
> > Most notably, the experiments are in what is now a relatively "toy"/old-school setting - training ResNets on CIFAR-10. While there are certainly some applications of ML that involve training similarly small models on similarly small datasets from scratch, the setup is a pretty far cry from current standard practice (which mainly involves fine-tuning much larger models).
>
> This is a good point. Indeed, it would be great to verify these findings in larger, more "practical" settings. Our intent was to focus on the early stage of training and in particular, before or around the point of linear mode connectivity onset where the observed phenomenon is most noticeable -- averaging seems to give much smaller gains later in training where learning rates are smaller as well. We believe that this would be difficult to do when finetuning larger models that are likely past the point of LMC onset, and instead, we choose to verify our observations in a relatively small-scale setting where we could feasibly perform multiple runs of an experiment.
>
> > I think the paper would be considerably stronger even with a small amount of additional experiments (e.g. with no augmentation, or with completely different data) and would be quite a lot stronger if it considered more experimental settings (say, language models, or sharpness-aware minimization).
>
> Thank you for the great suggestions! We agree with the action editor -- to this end, we add experiments with no augmentations, and child networks trained on non-overlapping splits of the dataset (before being averaged). These results can be found in Appendix D (specifically, Figures 10 and 11). We notice no significant differences in observations. One nuance we would like to point out is that the resulting interpolant is trained on the complete dataset in our experiments -- we see no obvious "correct" answer to this and feel that this is a reasonable choice.
>
> We also think that using different optimizers (or modifications to existing ones) in these experiments is a great direction for future work -- thank you!